# Differences in the Profile of Aromatic Metabolites in the Corresponding Blood Serum and Cerebrospinal Fluid Samples of Patients with Secondary Bacterial Meningitis

**DOI:** 10.3390/metabo15080527

**Published:** 2025-08-03

**Authors:** Alisa K. Pautova, Peter A. Meinarovich, Vladislav E. Zakharchenko, Pavel D. Sobolev, Natalia A. Burnakova, Natalia V. Beloborodova

**Affiliations:** 1Federal Research and Clinical Center of Intensive Care Medicine and Rehabilitology, Petrovka Street, 25-2, 107031 Moscow, Russia; pmeinarovich@fnkcrr.ru (P.A.M.); vzakharchenko@fnkcrr.ru (V.E.Z.); nbeloborodova@fnkcrr.ru (N.V.B.); 2Exacte Labs Bioanalytical Laboratory, 20-2 Nauchny Proezd, 117246 Moscow, Russia; pavel.sobolev@exactelabs.com (P.D.S.); n.burnakova@exactelabs.com (N.A.B.)

**Keywords:** 4-hydroxyphenyllactic acid, phenyllactic acid, indole-3-lactic acid, phenylpropionic acid, indole-3-propionic acid, nosocomial meningitis

## Abstract

**Background:** Secondary (nosocomial) bacterial meningitis remains a serious problem in patients with severe brain damage. The aim of this study was to assess the differences in the aromatic metabolites of tryptophan, phenylalanine, and tyrosine, in serum and cerebrospinal fluid (CSF) samples collected simultaneously from patients with long-term sequelae of severe brain damage with suspected secondary bacterial meningitis. **Methods:** Group I included 16 paired serum and CSF samples from patients (*N* = 11) without secondary bacterial meningitis; group II included 13 paired serum and CSF samples from patients (*N* = 4) with secondary bacterial meningitis. **Results:** The median concentrations of serum 5-hydroxyindole-3-acetic, CSF 4-hydroxyphenyllactic (*p*-HPhLA), CSF 4-hydroxyphenylacetic, CSF phenyllactic, and indole-3-lactic acids in serum and CSF were statistically higher in group II compared to group I (*p*-value ≤ 0.03), while 4-hydroxyphenylpropionic and indole-3-acetic in serum were lower in group II compared to group I (*p*-value = 0.04). In group I, *p*-HPhLA serum concentrations were greater than or equal to its CSF concentrations in 14 paired samples; in group II, *p*-HPhLA concentrations in serum were lower than in CSF in all paired samples. **Conclusions:** The obtained results demonstrate the differences in the profile of aromatic metabolites in serum and CSF and may confirm the hypothesis of the *p*-HPhLA microbial origin in the CSF of patients with secondary bacterial meningitis.

## 1. Introduction

Infectious complications remain a serious problem in intensive care units (ICUs) and surgical departments. Secondary (nosocomial) bacterial meningitis, also called healthcare-associated ventriculitis and meningitis, occupies a special place among infectious complications, with difficulties in verifying and timely diagnostics. Healthcare-associated ventriculitis and meningitis include infections associated with cerebrospinal fluid (CSF) shunts or drains, neurosurgery, head trauma, intrathecal drug therapy, and deep brain stimulation hardware [1,2]. In Russia, the incidence of secondary meningitis was evaluated as 9% with a 29% mortality rate [2]. This percentage is consistent with data about incidence and mortality in other countries (10% and 28%, respectively) [3]. In 2008 and then in 2024, the Centers for Disease Control (CDC) and Prevention proposed diagnostic criteria for central nervous system (CNS) infections, including meningitis or ventriculitis, that should contain at least one of several signs. To diagnose meningitis or ventriculitis, a positive CSF culture or PCR analysis result is sufficient but unnecessary [4,5]. This fact is certainly important, but it can lead to a false-positive result if the sample has been contaminated and has no signs of altered CSF composition. However, the main problem in diagnosing secondary meningitis is the lack of a positive CSF culture. This phenomenon may be due to preoperative antibiotic prophylaxis, treatment of postoperative complications not associated with CNS infections, and preanalytical factors [6,7]. A clinical presentation of the patient’s condition (fever and/or confused consciousness) or a change in CSF composition (increased leukocytes, elevated protein, and/or decreased glucose in the CSF) is, in fact, also extremely important. Considering these factors, in 2017, the Infectious Diseases Society of America proposed Clinical Practice Guidelines for Healthcare-Associated Ventriculitis and Meningitis [1]. In the case of neurosurgery or head trauma, a diagnosis of healthcare-associated ventriculitis or meningitis typically includes a positive CSF culture, increased leukocytes (leukocyte count more than 300 cells/mm^3^ and neutrophil predominance of more than 80%), and symptoms of infection, as well as decreased CSF glucose (less than 2.7 mmol/L) and elevated CSF protein concentrations (more than 1.0 g/L). However, several studies describe their non-specificity and ability to indicate, for example, tumors [1,8]. Some specific CSF tests can be used to confirm a bacterial diagnosis, and they are elevated CSF procalcitonin, an elevated CSF lactate, or a combination of both. However, for neonatal meningitis classification, CSF procalcitonin demonstrated a sensitivity of 82% and a specificity of 52% for diagnosing meningitis with an ROC-AUC = 0.7. Using a cut-off value of 2.07 mmol/L, CSF lactate showed a sensitivity of 84% and a specificity of 60% in meningitis diagnosis with an ROC-AUC of 0.7 [9]. In another study, the diagnostic ability of CSF lactate for secondary meningitis was reported. The sensitivity was 86% and the specificity was 100%, with an ROC-AUC = 0.98 and cut-off value of 4.95 mmol/L [10]. Thus, the overall classification ability of CSF procalcitonin and lactate was not perfect.

The CDC considers the development of serum diagnostic tests for bacterial meningitis a promising research direction. Lumbar puncture, although a standard medical procedure, carries some risks for patients and may lead to complications [11]. The investigation of specific serum biomarkers can make this manipulation not necessary for diagnostics. However, it may be challenging if other infection sites also exist. Serum markers, such as procalcitonin but not C-reactive protein, might point to healthcare-associated ventriculitis and meningitis after excluding other infection sites [12,13]. Thus, there are no reliable diagnostic markers in the blood in the case of secondary meningitis development against the background of some other infectious complications.

The search for new, more specific markers of secondary meningitis is a relevant research area [14,15,16]. The profiling of some serum metabolites in mice with a streptococcal meningitis model showed changes in regulating different pathways like cholic, lauric, and nicotinic acids [15]. One study involving untargeted CSF metabolomics profiling via NMR provided an etiologic classification of infectious agents in animals with meningitis [16]. An untargeted CSF metabolomics study on infants with or without bacterial meningitis was carried out using chromatography–mass spectrometry methods and resulted in nine metabolites (proline, N6-acetyllysine, taurine, cytidine, 2-hydroxyglutarate, ornithine, thymine, glutamate, and α-ketoglutarate), which were found to be the most important ones for separating patients into two groups, with an ROC-AUC= 0.97. Moreover, α-hydroxyisocaproic and 2-hydroxy-3-methylvaleric acids were higher, while sucrose was lower, in bacterial meningitis caused by *Streptococcus agalactiae* compared to all other cases of bacterial meningitis [17].

Untargeted analysis may be useful to understand relationships between concentrations of metabolites in serum and the CSF. The CSF and serum of donors (no data on their health was provided) generally had different metabolic profiles, with more considerable biological variation in the serum (coefficient of variation (CV): 26–180%) than in the CSF (CV: 17–130%). Only median concentrations of glycolic and glyceric acids were higher in the CSF than in the serum. For other hydroxyl acids, concentrations in the serum were 1.5–4 times higher than in the CSF, and for amino acids, they were 10 times higher than in the CSF [18]. There are also some clinical studies comparing the serum and CSF concentrations of different metabolites on patients with glioma [19] and patients with multiple sclerosis [20]. Such a comparative approach can simplify understanding the metabolite transport mechanisms through the blood–brain barrier (BBB) or the mechanisms of in situ metabolite production in the CNS.

A targeted search for potential microbial metabolites is the most justified. Serum concentrations of aromatic metabolites of amino acids, some of which are microbial metabolites, as shown in [21], have been shown to demonstrate prognostic and diagnostic significance in post-surgical and critically ill patients, including when monitoring over time [22,23,24]. Elevated CSF concentrations of 4-hydroxyphenyllactic acid (*p*-HPhLA) have been previously detected in post-neurosurgical patients with signs of secondary bacterial meningitis compared to those without signs of secondary bacterial meningitis [25]. Although elevated CSF concentrations of a proinflammatory interleukin-6 (IL-6) were also detected in post-neurosurgical patients with signs of secondary bacterial meningitis, subsequent ROC analysis together with analysis of the existing literature revealed its low specificity (sensitivity: 96% and specificity: 54%) in comparison with *p*-HPhLA (sensitivity: 67% and specificity: 83%) [25]. The low sensitivity for microbial *p*-HPhLA was attributed to the main limitation of that retrospective study. There were only eight cases of proven/suspected bacterial meningitis as an infectious complication reflected in medical documentation, including six positive bacterial CSF culture results, from a total of 30 patients with signs of post-neurosurgical meningitis. Thus, most patients were classified into the group with signs of post-neurosurgical meningitis based on non-specific criteria for CSF composition (glucose level less than 2.7 mmol/l; lactate level more than 4 mmol/l; protein level more than 1 g/l), and there could be false-positive classification results. Another limitation may be related to the lack of information on non-CNS infectious complications. We hypothesize that, on the one hand, *p*-HPhLA, the concentrations of which can be significantly increased in the serum of patients with infectious complications, has the ability to penetrate the BBB and lead to an increase in its CSF concentration. On the other hand, in the case of CNS infectious complications, the *p*-HPhLA CSF concentration can be higher than in serum. Thus, the absolute *p*-HPhLA CSF concentrations may have low information value. At the same time, the *p*-HPhLA ratio in the serum and CSF samples collected simultaneously from patients with suspected secondary bacterial meningitis may, on the contrary, provide more pathophysiological details.

The current study aimed to assess the ratio of *p*-HPhLA and other aromatic metabolites in serum and CSF samples collected simultaneously from patients with long-term sequelae of severe brain damage with suspected secondary bacterial meningitis, and to reveal if this ratio is specific for bacterial meningitis. To achieve this goal, we divided all patients involved in this study into two groups without (group I) and with culture-positive meningitis (group II). We also followed patients over time, since it is important to make a diagnostic decision about the presence or absence of meningitis at every point. This approach is fundamentally different from that used to predict the development of a disease, when only 1 point from a patient can be included in the study.

## 2. Materials and Methods

### 2.1. Study Desing

The study included patients (*N* = 15, 10 men, 5 women, aged from 21 to 82) with long-term sequelae of severe brain damage who were admitted to the ICUs at the Federal Research and Clinical Center of Intensive Care Medicine and Rehabilitology (Moscow, Russia). Approval from the Local Ethics Committee was obtained (N 3/24/3 from 13 November 2024) for the study at the Federal Research and Clinical Center of Intensive Care Medicine and Rehabilitology (Moscow, Russia). Informed consent was obtained from all subjects or their legal representatives involved in the study. Based on the medical documentation, patients were retrospectively divided into two groups according to the presence or absence of manifested secondary bacterial meningitis. Group I included patients without secondary bacterial meningitis (*N* = 11, 8 men, 3 women, aged 21 to 82); group II included patients with secondary bacterial meningitis (*N* = 4; 2 men and 2 women; aged 22 to 65). Most patients were studied over time (*N* = 8; 5 men and 3 women): two patients with meningitis had 2 points, one had 3 points, and one had 6 points. Three patients without meningitis had 2 points and one patient had 3 points. Four patients from group I had only one point. All necessary information about patients’ anamnesis, comorbidity, complications, treatment, clinical laboratory analyses of blood, and CSF were obtained retrospectively from medical documentation.

### 2.2. Blood Serum and Cerebrospinal Fluid Analysis

Blood (*n* = 29) and CSF (*n* = 29) samples were collected simultaneously within one day from patients (*N* = 15) with long-term sequelae of severe brain damage with suspected secondary bacterial meningitis. Blood samples were collected from the peripheral vein for routine laboratory analyses in an anticoagulant-free plastic vacuum tube. Its residues were centrifuged at 1500× *g* for 10 min. Then, serum was frozen and stored at −80 °C. CSF samples were collected for routine laboratory analyses in the case of suspected meningitis. Its residues were aliquoted into plastic Eppendorf tubes, frozen, and stored at −80 °C.

A complete blood count was carried out on a UniCel DxH800 (Beckman Coulter, Brea, CA, USA) hematology analyzer; blood and CSF biochemistry tests were performed using an AU480 chemistry analyzer (Beckman Coulter, Brea, CA, USA). Leukocyte count and leukocyte ratios were measured on an Olympus CX41 light microscope (Olympus, Tokyo, Japan). A cultural method was used to define microorganisms in biological samples. All these standard analyses were performed in a clinical laboratory at the Federal Research and Clinical Center of Intensive Care Medicine and Rehabilitology (Moscow, Russia).

All biological samples were subsequently analyzed using ultra-high-pressure liquid chromatography–tandem mass spectrometry (UPLC-MS/MS): the SCIEX ExionLC AC System (AB Sciex, Framingham, MA, USA) with a reverse-phase column Waters Acquity UPLC HSS C18 (50 mm × 2.1 mm, 1.7 µm) (Waters Corporation, Milford, MA, USA) coupled with an AB Sciex Triple Quad 5500 Plus mass spectrometer (AB Sciex, Framingham, MA, USA). Aromatic metabolites of phenylalanine (3-phenylpropionic acid—PhPA, 3-phenyllactic acid—PhLA), tyrosine (4-hydroxybenzoic acid—*p*-HBA, 4-hydroxyphenylacetic acid—*p*-HPhAA, 4-hydroxyphenyllactic acid—*p*-HPhLA, 4-hydroxyphenylpropionic acid—*p*-HPhPA), and tryptophan (indole-3-carboxylic acid—3ICA, indole-3-propionic acid—3IPA, indole-3-acetic acid—3IAA, 5-hydroxyindole-3-acetic acid—5HIAA, indole-3-lactic acid—3ILA) with indole-3-acetic acid-d_4_ as an internal standard were determined using protocols which were validated according to the Food and Drug Administration (FDA) Guidance for Industry “Bioanalytical Method Validation”, May 2018 [26], and a guideline by the International Council for Harmonization (ICH) on bioanalytical method validation [27]. Two different sample preparation protocols were applied to determine aromatic metabolites at nmol/L and μmol/L concentration levels. For the serum analysis of all metabolites, except *p*-HPhPA and 3ICA, the sample preparation protocol included protein precipitation and subsequent UPLC-MS/MS analysis (μmol/L concentration levels) [28]; for the CSF and serum analysis of *p*-HPhPA and 3ICA, the sample preparation protocol included protein precipitation, sample concentration and subsequent UPLC-MS/MS analysis (nmol/L concentration levels) [29]. The CV for the determination of the analyte concentrations did not exceed 15%. For ease of data comparison in the text, all data on concentrations are presented in nmol/L.

### 2.3. Statistical Analysis and Models

All data analysis, statistics, and visualization were performed with Python 3.0 libraries (Pandas 2.3.1, SciPy 1.16.0, Matplotlib 3.10.5, Seaborn 0.13.2, Plotly 6.2.0, and Sklearn1.7.1). Clinical, biochemical, and metabolomic data were described by median, interquartile range (for quantitative variables), and number of cases (for discrete variables). For statistical analysis, we changed to 0 data describing concentrations of metabolites that were less than the lower limits of quantitation. All data on blood and CSF clinical and biochemical analyses and concentrations of aromatic metabolites in the blood serum and CSF samples of patients with long-term sequelae of severe brain damage are collated in Appendix A. In the tables of the Section 3, we specify if values are lower than the respective lower limit of quantitation.

To account for the sample belonging to a specific patient, a mixed-effects model was used. The two groups of samples in tables of Section 3.2 and Section 3.3 were compared using the Wald test for fixed effects. The significance level was chosen to be a *p*-value of less than 0.05. For multiple pairwise comparisons, the Bonferroni correction was applied. For all the tests performed, MixedLM from the Statsmodels 0.14.5 package for Python 3.0 was used.

Due to the small number of patients and the relatively large number of points in the dynamics from these patients included in the study, the objectivity of constructing prognostic models is disputable. Despite this, we conducted this analysis, but its results are presented in Appendix A. This data analysis included transformation with Standard Scaler. Principal component analysis (PCA) using the serum and CSF concentrations of metabolites was performed to visualize and evaluate the linear separability of the samples. The percentage of explained variance was computed. Loading plots demonstrated the impact of every feature in each component.

A machine learning approach was performed with the aim of evaluating the ability of metabolomics data to discriminate patients by group. For an explanation of the feature importance, we used the coefficients of a support vector machine classifier (SVC, Sklearn [30]). The linear SVC is a method for building models on small datasets, although it does not consider nonlinear interactions [31]. We chose a Boruta algorithm for the feature selection [32]. It compares the impact of every feature with the impact of the added noisy variables. The features that were selected by the Boruta algorithm were then used for the pipeline fitting. It included a normalization (Standard Scaler) and SVC with a linear kernel. The cross-validation (train/test = 5/1, k = 10) was performed to compute importances (the linear model coefficients for every feature) [33].

To explain the diagnostic significance and find the cut-off values, receiver operating characteristic (ROC) analysis was performed for important parameters and for the overall model (the coefficients of the SVC models were multiplied on the corresponding features to receive the summary features). ROC curves were plotted in the true positive rate (TPR, sensitivity) and false positive rate (FPR, 1-specificity) coordinates. In addition, 95% confidence intervals (CIs) were computed using the bootstrap technique (*n* = 100) [33]. Cut-off values were calculated through Jouden’s index.

## 3. Results

### 3.1. Patients with Long-Term Sequelae of Severe Brain Damage

Group I included eleven patients without secondary bacterial meningitis (Table 1). The primary diagnoses included stroke (*N* = 6), cerebral hematoma (*N* = 7), and traumatic brain injury (*N* = 7). Concomitant diseases included diabetes mellitus (*N* = 2), hypertension (*N* = 8), ischemic heart disease (*N* = 8), and gastrointestinal disorders (*N* = 8). Some patients had invasive monitoring of intracranial pressure (*N* = 6) and ventriculoperitoneal shunt (*N* = 5). Decompressive trepanation was performed in six patients. Infectious diseases in group I were explained by nosocomial pneumonia (*N* = 8) and urogenital infections (*N* = 5). Mechanical ventilation and tracheostomy were used in eight patients. Ten patients had depressed consciousness, and eight had a fever (t > 38 °C).

Group II included four patients with secondary bacterial meningitis. *Klebsiella pneumonia* was detected in all CSF samples of these patients. One patient also had *Candida parapsilosis* in the CSF. The primary diagnoses included stroke (*N* = 2), cerebral hematoma (*N* = 2), and traumatic brain injury (*N* = 1). Concomitant diseases included diabetes mellitus (*N* = 1), hypertension (*N* = 3), ischemic heart disease (*N* = 3), and gastrointestinal disorders (*N* = 2). Some patients also underwent invasive intracranial pressure monitoring (*N* = 3) and ventriculoperitoneal shunts (*N* = 1). Decompressive trephination was performed in one patient. All patients from group II had nosocomial pneumonia; two patients had urogenital infections. Mechanical ventilation was used in three patients. Two patients had depressed consciousness, and four had a fever (t > 38 °C).

Antibacterial therapy included the usage of meropenem (1000 mg 3 times per day) or cefoperazone + sulbactam (2 g + 2 g 2 times per day) in group I and meropenem (1000 mg 3 times per day), imipenem + cilastatin (0.5 g + 0.5 g 4 times per day), linezolid (600 mg 2 times per day), amikacin (1.5 g 1 time per day), and vancomycin (1000 mg 2 times per day) in group II (depending on the type of pathogen). In addition, all these patients receive rifaximin (200 mg 2 times per day) enterally with the aim of gut decontamination. One patient from group II had fluconazole (400 mg 1 times per day) because of *Candida parapsilosis* in CSF.

All patients from group II died within 30 days from the last sample collection. In contrast, all patients from group I survived and were successfully discharged from the ICU.

### 3.2. Blood Serum and Cerebrospinal Fluid Clinical and Biochemical Analysis

Most patients (8 of 15) were studied in terms of dynamics; the total number of samples was 16 in group I without secondary bacterial meningitis and 13 in group II with secondary bacterial meningitis. Since the main idea of our study was to reveal the differences between the CSF and serum samples of patients with or without meningitis, we studied all points of patients independently, and each time, we asked whether there was meningitis in the new point or not. Thus, theoretically, samples from one patient could have been classified into different groups over time, but this did not happen in our study and all samples from the same patients always remained in the same groups. All blood and CSF samples of patients in groups I and II were analyzed to reveal a number of clinical and biochemical parameters (Table 2).

To account for the sample belonging to a specific patient, a mixed-effects model (Wald test) was used for the comparison of two groups. Statistically significant differences between serum samples were found in hemoglobin, hematocrit, total protein, glucose, albumin, and urea; urea, in most cases, was within reference values. All these parameters are non-specific for infectious complications, particularly for secondary meningitis. Although the commonly known infectious markers like leukocyte count or C-reactive protein were higher than their reference values, they were not statistically different between blood samples in the two patients’ groups.

Regarding CSF composition, statistically significant changes were detected in the relative number of neutrophils and lymphocytes, and protein. Parameters that are commonly used in secondary meningitis diagnosis are a leukocyte count of more than 300 cells/mm^3^ with a relative number of neutrophils higher than 80%, glucose less than 2.7 mmol/L, and proteins more than 1.0 g/L. Priority for secondary meningitis is usually given to neutrophilic pleocytosis. Regarding group I, neutrophilic pleocytosis was not detected in any cases; group II with secondary bacterial meningitis was characterized by four CSF samples (31%) without pleocytosis and two samples with a neutrophil count less than 80%. Glucose was both above (*n* = 11 for group I and *n* = 4 for group II) and below (*n* = 5 for group I and *n* = 9 for group II) 2.7 mmol/L, as well as protein (*n* = 5 for group I and *n* = 10 for group II) and below (*n* = 11 for group I and *n* = 3 for group II) 1.0 g/L in sample groups.

### 3.3. Blood Serum and Cerebrospinal Fluid Aromatic Metabolites

All serum and CSF samples were analyzed using UPLC-MS/MS to determine 11 aromatic metabolites of phenylalanine, tyrosine, and tryptophan. *p*-HPhLA, *p*-HBA, *p*-HPhAA, PhLA, and 3ICA were detected in all serum and CSF samples; 5HIAA, 3ILA, and 3IAA were detected in most serum and CSF samples; 3IPA and *p*-HPhPA were detected in less than half of the cases; and PhPA was detected in only one serum sample.

Reference values in serum for these metabolites were previously obtained by analyzing the serum samples of healthy donors [28,29]. These data allowed us to conduct a comparative analysis and find out which of the metabolites in the patients’ groups were significantly different from the norm (Table 3). All metabolites were statistically different between donors and group I, except for *p*-HPhLA and 5HIAA. Similarly, all metabolites were statistically different between donors and group II, except for 5HIAA, 3ILA, and *p*-HPhPA. In addition, healthy donors are characterized by PhPA and 3IPA, while these metabolites were mostly not detected in patients’ groups.

In Figure 1, we can see that samples from groups I (green points) and II (red points) can be clearly classified using the 3IAA and 3ILA serum concentrations. At the same time, healthy donors and patients without meningitis (group I) are not clearly classified by 3IAA and 3ILA serum concentrations. However, this can be achieved additionally using 3IPA in serum that is high in healthy donors and was not quantitatively measured in most samples of patients from groups I and II (Table 3).

Reference values for metabolites of interest in CSF are currently unavailable to us due to the lack of CSF from healthy people for ethical reasons and the rather scanty literature data on this matter. In this regard, we had to limit ourselves to only a comparative analysis of metabolite concentrations in two groups of samples (Table 4). It can be noted that we duplicated the patients’ serum metabolite concentrations from Table 3 in Table 4 to assess differences in serum and CSF concentrations of the same metabolite in the same group of samples. Median concentrations of 3ILA in serum and CSF were statistically higher in group II compared to group I (*p* ≤ 0.02). Median concentrations of *p*-HPhLA, *p*-HPhAA, and PhLA in CSF and 5HIAA and 3IAA in serum were statistically higher in group II compared to group I (*p* ≤ 0.03), while median concentrations of 3IAA and *p*-HPhPA in serum were statistically higher in group I compared to group II (*p* = 0.04).

Boxplots (Figure 2) were constructed for parameters that were found to be statistically significantly different in Table 3 to illustrate the distribution of metabolite concentrations for each patients’ group, while the colored dots represent individual data points for each subject, with each color corresponding to a different patient. Two metabolites, PhLA and *p*-HPhLA in CSF, show clear, non-overlapping differences between groups, highlighted by a shaded orange band indicating the range where the groups do not overlap, suggesting strong discriminatory potential. Other metabolites display substantial overlap between groups. Overall, this figure provides a comparative overview of metabolite concentrations, enabling the assessment of both group-level trends and individual variability, with highlighted separation zones pointing to metabolites that may serve as strong discriminators between the two groups.

Regarding the aromatic metabolite ratio in the paired serum and CSF samples, serum concentrations of all metabolites, except *p*-HPhLA and 5HIAA, were higher than in the CSF. We also present these results in Figure 3 using *p*-HPhLA and 5HIAA serum concentrations on the X-axis, and *p*-HPhLA and 5HIAA CSF concentrations on the Y-axis. In group I, *p*-HPhLA serum concentrations were greater than or equal to CSF concentrations in most (*n* = 14) samples (green dots); in group II, *p*-HPhLA concentrations in serum were lesser than in CSF in all samples (Figure 3a, red dots). Thus, in the case of *p*-HPhLA, the samples in groups I and II are separated by the y = x blue line. In group I, 5HIAA serum concentrations were greater than CSF concentrations in 6 samples (green dots); in group II, 5HIAA concentrations in serum were lower than in CSF in 11 samples (Figure 3b, red dots). Thus, in the case of 5HIAA, samples from both groups are located randomly relative to the y = x blue line.

We conducted a correlation analysis of all metabolites in the two groups of samples (Figure 4). The aim of this analysis was to reveal if there was a relationship between the concentrations of the same metabolites in serum and CSF, as well as to identify other potentially interesting relationships:(1)In group I without secondary meningitis, concentrations of the same metabolites in the serum and CSF correlate significantly and strongly for *p*-HBA (r = 0.82), *p*-HPhAA (r = 0.75), 3ICA (r = 0.80), and 3IPA (r = 0.70); and significantly and moderately for *p*-HPhPA (r = 0.56) and PhLA (r = 0.58). It is noteworthy that such a correlation was not found for *p*-HPhLA. In group II with secondary meningitis, the concentration of only *p*-HPhAA in the serum and CSF correlates significantly and strongly (r = 0.87).(2)In serum samples from group I, there were no strong and significant correlations. In serum samples from group II, there were strong negative and significant correlations between 3ILA and *p*-HBA (r = −0.75) and strong positive and significant correlations between PhLA and *p*-HPhLA (r = 0.74), *p*-HPhPA and *p*-HPhAA (r = 0.76), 3IPA and *p*-HPhPA (r = 0.77), and 3IAA and 3ILA (r = 0.82).(3)In CSF samples from group I, there were strong and significant correlations between *p*-HPhAA and 3ILA (r = 0.84) and 3IAA (r = 0.75); 3IAA and 3ILA (r = 0.74); and a significant and moderate correlation between 3IAA and 3IPA (r = 0.51). In CSF samples from group II, there were strong and significant correlations between *p*-HPhLA and 3ICA (r = 0.79); 3ILA and PhLA (r = 0.76); 3IAA and 5HIAA (r = 0.75), and 3ILA (r = 0.88); 3IPA and *p*-HPhAA (r = 0.74), 3ILA (r = 0.72), and 3IAA (r = 0.73); and a significant and moderate correlation between *p*-HPhAA and 3IAA (r = 0.63).

## 4. Discussion

Our study assessed the content and ratio of aromatic metabolites in the serum and CSF samples collected simultaneously from patients with long-term sequelae of severe brain damage and suspected secondary bacterial meningitis. This pilot study aims to identify some novel and previously undescribed patterns of aromatic amino acid metabolites in the serum and CSF of patients with secondary CNS infections.

Patients with long-term sequelae of severe brain damage were included in our study. These patients are also called patients with a chronic critical illness [34]. This group of patients is characterized by an extended stay in the ICU, in most cases on prolonged mechanical ventilation. Unfortunately, the development of nosocomial infections (pneumonia, secondary meningitis, ventriculitis, urogenital tract infections, and soft tissue infections) is frequent in these patients. Nosocomial pneumonia caused by *Klebsiella pneumoniae* occupies a leading position in these patients [35]. In our study, most patients (12 out of 15 patients, 80%) had nosocomial pneumonia caused by *Klebsiella pneumoniae* (Table 1).

Since the initial diagnosis of our group of chronic critically ill patients was associated with severe brain damage of various origins, the development of a secondary nosocomial CNS infection, namely secondary bacterial meningitis, was a common complication. We only chose patients with culture-confirmed secondary meningitis to avoid confounding factors related to culture-unconfirmed secondary meningitis. However, culture-confirmed secondary meningitis is quite rare because most patients have received prior antimicrobial therapy [6], which can lead to false-negative culture results. In particular, patients with a chronic critical illness may have more than one episode of antimicrobial treatment in their anamnesis. Thus, group II with secondary bacterial meningitis included only four patients.

All of these secondary bacterial meningitis cases were also caused by nosocomial *Klebsiella pneumoniae*. This pathogen has been reported to be the second most frequently detected cause of secondary bacterial meningitis in patients with traumatic brain injury admitted into neuro-critical care units [36]. In our study, *Klebsiella pneumoniae* in CSF may be due to recent brain surgery or implanted invasive devices (intracranial pressure monitoring or ventriculoperitoneal shunts).

Despite the small number of patients with secondary bacterial meningitis included in group II, we studied their paired serum and CSF samples in terms of dynamics. We have previously demonstrated that serum concentrations of aromatic metabolites can be successfully used for monitoring critically ill patients’ condition during extracorporeal blood purification [23]. We suspected that the information at each sample collection point in our current study could be used independently to confirm the presence or absence of an infectious process. Such an approach could help us to notice positive or negative dynamics in a patient’s condition, i.e., resolved or newly developed meningitis. However, in our pilot study, such situations were not observed, and all samples from group I without secondary meningitis at all time points were classified as samples without meningitis; similarly, all samples from group II with meningitis at all time points were characterized as samples with meningitis.

During the clinical and biochemical analysis of serum samples in groups I and II, a number of parameters were identified as statistically different between the two groups (Table 2). At the same time, they had low clinical significance for diagnosing secondary meningitis. The potentially significant infectious markers, i.e., leucocyte count or C-reactive protein, were higher than their reference values, indicating the systemic inflammatory process mainly caused by nosocomial pneumonia. However, they did not differ between the two groups and cannot be used for the differential diagnosis of meningitis.

Some potentially significant parameters in the clinical and biochemical analysis of CSF were statistically different between the two groups, including the relative number of neutrophils and lymphocytes, and protein (Table 2). However, no parameters among them were group-specific. The obtained results highlight the need to search for new, more specific markers for secondary meningitis diagnosis.

Our research group has been studying the content of phenyl-containing aromatic metabolites in various groups of patients for a long time. The concentration of some metabolites of phenylalanine and tyrosine in blood serum increases by tens of times in patients with a systemic infectious process (sepsis) [22] or only a few times in those with local infectious complications (pneumonia and other postoperative complications) [24] in comparison with healthy donors, and it has statistical significance in the prognosis of the development of infectious complications and patient outcomes. These metabolites are PhLA, *p*-HPhAA, and *p*-HPhLA, also called sepsis-associated microbial metabolites. We conducted a pilot clinical study on chronic critically ill patients and found that the concentrations of our metabolites in these patients were generally not significantly different from healthy donors, despite statistical differences in the number of metabolites [37]. In our current study, the serum profiles of phenyl-containing aromatic metabolites in patients from groups I and II were statistically different from those in healthy donors, except *p*-HPhLA in group I. Our hypothesis for the observed differences between our group of chronic critically ill patients and the previously described group of patients with a similar condition is the fact that secondary meningitis was suspected in our patients. Despite the absence of documented meningitis in group I, the condition of these patients caused concern among the treating physicians.

The serum PhLA in group II with a median concentration of 5602 nmol/L corresponded to that concentration (4000 nmol/L) in critically ill patients admitted to the ICU, who subsequently died [22]. Notably, all patients in group II died within 30 days. Thus, we can assume that serum PhLA is a prognostic marker of mortality both in our patients with secondary meningitis and in critically ill patients on admission to the ICU, and we will continue research in this direction.

Regarding the serum profile of indole-containing metabolites, we currently have little data to help interpret the findings. According to the literature and data accumulated in the open internet resource KEGG PATHWAY Database [38], tryptophan can be metabolized with or without the opening of the indole ring. The latter is called the indole pathway and can then proceed via the serotonin pathway to form 5HIAA and other neurotransmitters or via the indole-3-pyruvate pathway, which is considered microbial, with further metabolism to 3IAA, 3IPA, 3ILA, and 3ICA [39,40,41,42,43].

It can be noted that, as in the case of phenyl-containing metabolites, particularly PhPA, there are two metabolites, namely 3IAA and 3IPA, whose levels were reduced compared to healthy donors. Serum PhPA at constant low concentrations (approximately 500 nmol/L) is known to characterize healthy people, and in patients, we usually record its disappearance (when it reaches a concentration less than the lower limit of quantitation) [22]. This may be related to the fact that PhPA derivatives are considered anti-inflammatory substances and decrease the release of proinflammatory IL-1b through influence on the NLRP3 inflammasome [44].

Serum 3IPA at constant low concentrations may also characterize healthy people. Dodd et al. demonstrated that gnotobiotic mice colonized with *C. sporogenes* with knockout gene fldC had low concentrations of propanoic acids (3IPA, PhPA, and *p*-HPhPA). This fact resulted in elevated permeability of the gut barrier and impaired immune status as a hyperactivated response to microorganisms [21]. While 3IPA negatively correlated with the serum levels of many proinflammatory mediators in patients with acute COVID-19, 3IAA and 3ILA levels positively correlated with levels of acute-phase proteins and proinflammatory cytokines [45]. Similarly to PhPA, we recorded its disappearance in the serum samples of our patients (Table 3). However, serum 3IAA was decreased in patients from both groups.

With respect to 3IAA, there are divergences in opinion in the literature. 3IAA can disrupt the metabolism of gut secretory cells and is associated with the influence of stress [46]. Oppositely, the anti-inflammatory effects of this acid were reported to be carried out through AhR-related mechanisms [47]. This could be related to the fact that the oxidation of tryptophan through indole-3-pyruvate to 3IAA is not implemented with participation of the porA gene as it occurs during PhAA and *p*-HPhAA synthesis [21,42].

The concentrations of serum 3ILA in the healthy donors were in the range between those of groups I and II, which is difficult to interpret and certainly requires further studies in a larger group of patients and patients with other types of infectious complications. In the literature, 3ILA in feces was associated with Bifidobacterium-dominated microbiota and could decrease inflammation in intestinal epithelial cells [48].

A combination of serum 3ILA, 3IAA, and 3IPA resulted in a very illustrative separation of groups I and II and healthy donors, which is demonstrated in Figure 1.

5HIAA is a well-known serotonin metabolite and its serum concentration is a potential indicator for the diagnosis and monitoring of carcinoid tumors originating from enterochromaffin cells in the small intestine; hence, it is not usually considered a marker of infectious complications [49]. Although its concentrations in serum were statistically different between groups I and II, no statistical differences were observed between the healthy donors and patients from groups I and II, demonstrating that all differences were within normal values. Therefore, this metabolite has no diagnostic value for secondary meningitis.

CSF concentrations of phenyl-containing acids, particularly sepsis-associated PhLA, *p*-HPhAA, and *p*-HPhLA, were statistically higher by more than 10 times in group II with secondary meningitis than in group I (Table 4). In our previous study on post-neurosurgical meningitis, the sensitivity of the GC-MS method was not enough to measure these three metabolites in all CSF samples (only *p*-HPhLA was detected in all CSF samples) [25]. Thus, using UPLC-MS/MS in the current study allowed us to obtain very important data, which demonstrated similar changes in the profile of sepsis-associated metabolites in blood serum [22] and CSF in the case of severe infectious complications that could lead to the patient’s death.

The median CSF concentrations of PhLA in group II were 11 times higher than in group I. However, its serum concentrations were higher than in the corresponding CSF samples. If PhLA can penetrate the BBB from blood, higher CSF concentrations of PhLA could reflect the more severe state of patients from group II but not secondary meningitis. This fact can be indirectly confirmed by the presence of a relatively low (r = 0.58), but still statistically significant, correlation between serum and CSF PhLA in group II with secondary meningitis (Figure 3), while there was no correlation between the same parameters in group I without secondary meningitis.

3ILA concentrations in CSF were statistically higher in group II than in group I (Table 4). Since there is almost no research in the literature on tryptophan microbial metabolites in CSF in cases of CNS infectious complications, we can assume that the presence of 3ILA may reflect the microbial metabolism of tryptophan in the CNS in the case of secondary meningitis in patients from group II. Additionally, we found no correlations between the serum and CSF concentrations of 3ILA in patients from group II (Figure 4), which could reflect that this metabolite did not penetrate the BBB from blood, where its serum concentrations were higher than in CSF (Table 4).

5HIAA in CSF deserves special attention since it is not considered a microbial metabolite and has a known mechanism of quick formation using mitochondrial enzyme monoamine oxidase from the neurotransmitter serotonin, which is synthesized in the raphe nuclei of the brain [49]. Its direct formation in the CNS explains the observed cases when 5HIAA concentrations in CSF were higher than in serum (Figure 3b). However, these results were not specific for groups I or II and, thus, did not reflect infectious processes.

Our study was conducted to prove the hypothesis of the *p*-HPhLA accumulation mechanisms in CSF, which had been postulated previously [25] and was briefly formulated in the Section 1. We can state that we have managed to confirm this hypothesis. In the case of secondary bacterial meningitis, *p*-HPhLA could be formed directly in the CNS since we detected higher CSF concentrations of it compared to those in serum in all samples from group II (Figure 3a). Its statistically higher CSF concentrations in patients with secondary meningitis in group II also confirm this hypothesis. We achieved these results because we analyzed only culture-confirmed secondary meningitis.

The second part of the hypothesis about *p*-HPhLA accumulation in the CSF included possible *p*-HPhLA penetration from the bloodstream through the BBB. *p*-HPhLA is a hydrophilic compound with a hydroxyl and a carboxyl group, and accordingly, low lipophilicity, which complicates passive diffusion through the lipid layer of the BBB membranes, and it also has a relatively high molecular weight of 182 Da. The most possible routes of its penetration through the BBB are via (1) the LAT1 L-type amino acid transporter 1 that is used for tyrosine, phenylalanine, and tyrosine transport [50], and (2) OAT organic anion transporters [51]. Thus, in group I with pneumonia, in which the *p*-HPhLA concentrations in the blood serum were higher than or equal to the *p*-HPhLA concentrations in the CSF (Table 4), such active mechanisms could be assumed. However, the lack of correlation between the *p*-HPhLA concentrations in the serum and CSF of this group of patients (Figure 4) does not allow us to confirm this hypothesis at this time.

## 5. Conclusions

This study aimed to determine the accumulation possibility of a clinically significant sepsis-associated metabolite, *p*-HPhLA, in the CSF of patients with long-term sequelae of severe brain damage with secondary CNS infection. We found that the CSF concentrations of *p*-HPhLA in patients with culture-positive secondary meningitis were higher than in the serum samples of the same patients, which may prove our hypothesis on *p*-HPhLA’s direct microbial origin in the CNS. Since we also detected lower *p*-HPhLA concentrations in the CSF of patients without secondary meningitis than in their serum, we supposed that *p*-HPhLA could penetrate the BBB. However, we did not confirm this hypothesis because we did not obtain a statistically significant correlation between the CSF and serum concentrations of *p*-HPhLA. We proposed experiments with isotopically labeled *p*-HPhLA analogs to prove or disprove this hypothesis. In addition, we still do not have data on the normal concentrations of *p*-HPhLA in CSF, but we will work in this direction in the future.

The profiles of other potentially significant aromatic metabolites were found to be altered in the serum and CSF samples of patients from both groups. However, we have little data on their concentrations in patients with other types of infectious complications and will continue our research to obtain more specific clinical data.

## Figures and Tables

**Figure 1 metabolites-15-00527-f001:**
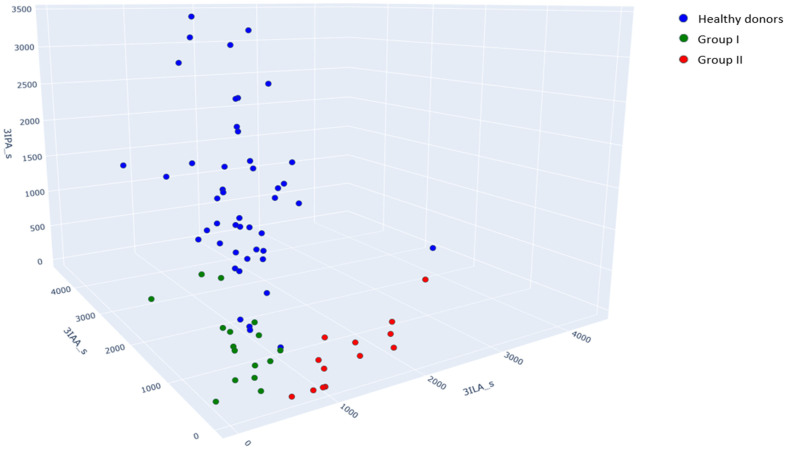
Scatter plot with absolute values of the 3ILA, 3IAA, and 3IPA serum concentrations, nmol/L. Green points correspond to samples from group I (patients without secondary bacterial meningitis), red points correspond to samples from group II (patients with secondary bacterial meningitis), and blue points correspond to healthy donors’ samples.

**Figure 2 metabolites-15-00527-f002:**
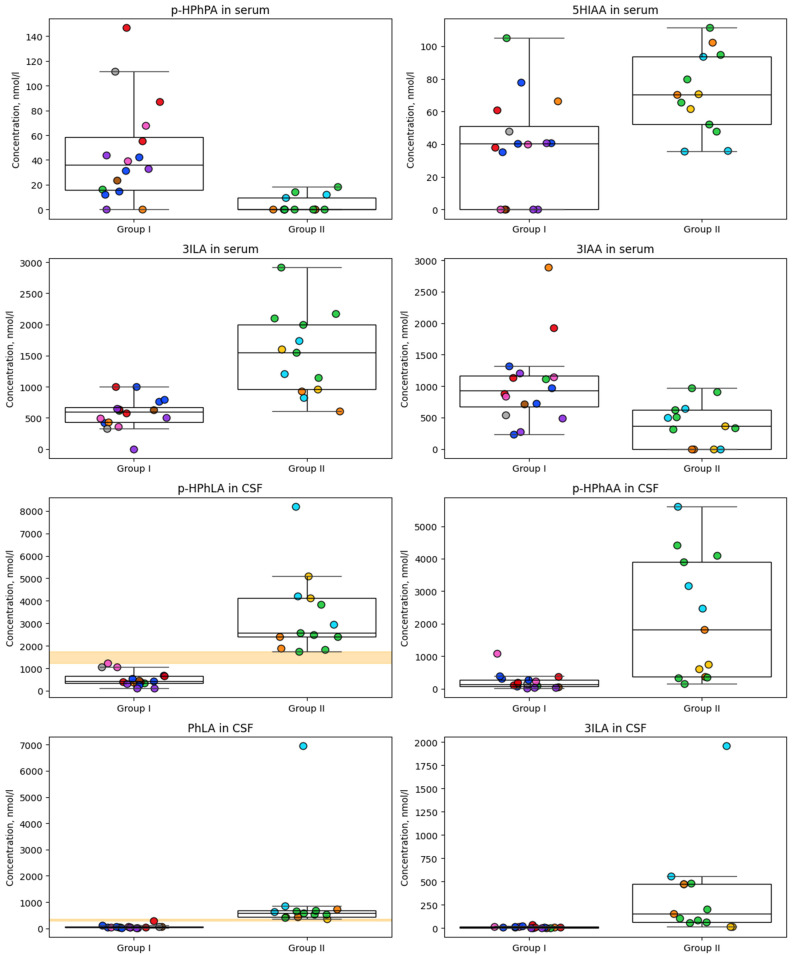
Concentrations (nmol/L) of metabolites in the cerebrospinal fluid and serum samples of patients without secondary meningitis (group I) and with secondary meningitis (group II) that differ significantly between groups (Table 4). The light orange zone for *p*-HPhLA and PhLA in the CSF explains the gap between groups.

**Figure 3 metabolites-15-00527-f003:**
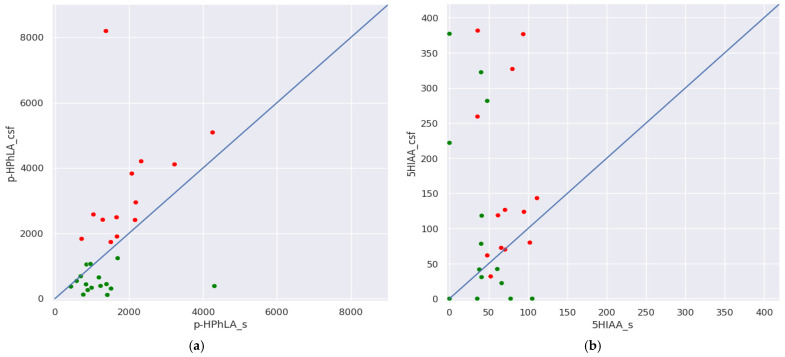
Concentrations (nmol/L) of (**a**) 4-hydroxyphenyllactic (*p*-HPhLA) and (**b**) 5-hydroxyindole-3-acetic (5HIAA) acid in the cerebrospinal fluid and serum samples of patients without secondary meningitis (group I, green dots) and with secondary meningitis (group II, red dots). Blue line explains equality of concentrations (y = x).

**Figure 4 metabolites-15-00527-f004:**
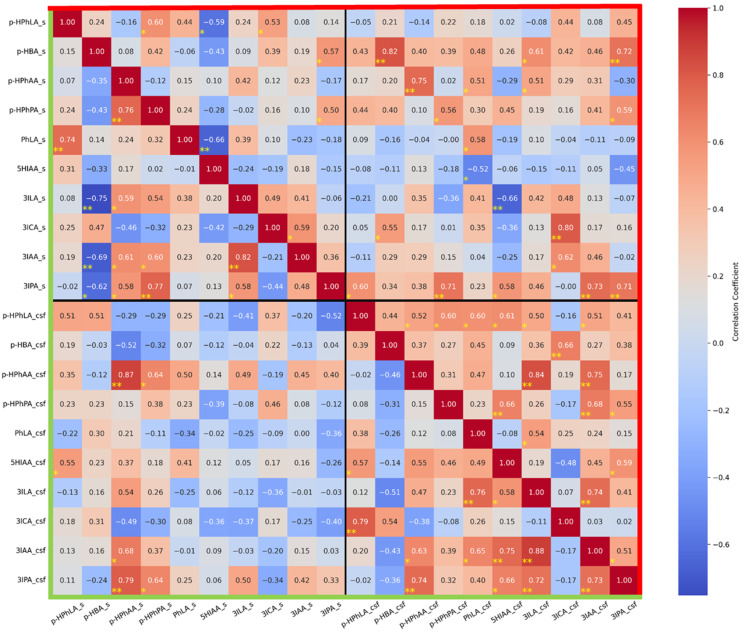
A correlation matrix for concentrations of metabolites in serum and CSF between samples from group I (patients without secondary bacterial meningitis) and samples from group II (patients with secondary bacterial meningitis). The red corner (upper right) corresponds to pairwise correlations for group I, while the green corner (lower left) corresponds to group II. Statistically significant values are marked with one or two yellow asterisks (*p* < 0.05 and *p* < 0.01, respectively).

**Table 1 metabolites-15-00527-t001:** Description of patients with long-term sequelae of severe brain damage.

Parameter	Group I. Patients Without Secondary Bacterial Meningitis (*N* = 11)	Group II. Patients with Secondary Bacterial Meningitis (*N* = 4)
Age, years	55 (38, 62)	58 (45, 64)
Sex	8 men, 3 women	2 men, 2 women
Length of stay at the time of sample collection, days	17 (13, 24)	31 (16, 60)
Depressed consciousness	10	2
Elevated body temperature, >38 °C	8	4
Deaths	0	4
**Primary Diagnoses**		
Stroke	6	2
Cerebral hematoma	7	2
Traumatic brain injury	7	1
Stroke	6	2
**Surgical Risk Factors of Bacterial Meningitis**		
Ventriculoperitoneal shunt	5	1
Invasive monitoring of intracranial pressure	6	3
Decompressive trepanation	6	1
Mechanical ventilation and tracheostomy	8	3
**Concomitant Diseases**		
Diabetes	2	1
Hypertension	8	3
Ischemic heart disease	8	3
Gastrointestinal disorders	8	2
**Infectious Complications**		
Meningitis	0	4
Pneumonia	8	4
Urogenital tract infections	5	2
**Results of Microbiological Analysis**		
CSF	*Staphylococcus epidermidis*: 3	*Klebsiella pneumonia*: 4 *Candida parapsilosis*: 1
Bronchoalveolar lavage	*Klebsiella pneumonia*: 7 *Pseudomonas* spp.: 1 *Acinetobacter* spp.: 2 *Stenotrophomonas maltophilia*: 1	*Klebsiella pneumonia*: 2 *Acinetobacter* spp.: 2
Blood	*Staphylococcus* spp.: 3 *Acinetobacter* spp.: 1	*Klebsiella pneumonia*: 1
Urine	*Klebsiella pneumonia*: 1 *Candida albicans*: 6 *Escherichia coli*: 8	*Candida albicans*: 1 *Proteus* spp.: 1

**Table 2 metabolites-15-00527-t002:** Results of blood and cerebrospinal fluid clinical and biochemical analyses of patients with long-term sequelae of severe brain damage.

Parameter	Reference Value	Group I. Samples (*n* = 16) from Patients Without Secondary Bacterial Meningitis	Group II. Samples (*n* = 13) from Patients with Secondary Bacterial Meningitis	*p*-Value *
**Blood**				
Leukocytes, 10^9^	4.0–9.0	10.6 (7.8, 11.4)	11.5 (8.6, 16.1)	0.25
Hemoglobin, g/L	130–160	119.5 (100.2, 129.2)	90.0 (81.0, 104.0)	**0.02**
Hematocrit, %	35.0–50.0	35.2 (29.9, 38.2)	28.8 (24.1, 31.7)	**0.03**
Platelets, 10^9^	180–320	286 (255, 337)	194 (147, 276)	0.25
Total protein, g/L	66.0–88.0	61.0 (54.9, 65.6)	51.1 (50.4, 52.2)	**0.03**
Glucose, mmol/L	3.9–6.4	5.8 (5.6, 6.2)	7.6 (5.4, 10.1)	**0.03**
Albumin, g/L	34.0–50.0	35.8 (32.8, 38.5)	28.7 (25.5, 31.2)	**0.03**
Creatinine, μmol/L	53.0–115.0	72.3 (55.2, 80.7)	53.9 (49.8, 66.4)	0.25
Urea, mmol/L	3.0–9.2	4.3 (3.1, 5.1)	6.5 (5.2, 12.1)	**0.01**
C-reactive protein, mg/L	0.0–5.0	21.1 (0.7, 73.6)	31.6 (21.2, 66.5)	0.49
International normalized ratio	0.8–1.2	10.6 (7.8, 11.4)	11.5 (8.6, 16.1)	0.75
Activated partial thromboplastin time, sec	25.4–36.9	119.5 (100.2, 129.2)	90.0 (81.0, 104.0)	0.05
**Cerebrospinal Fluid**				
Leukocyte count, cells/mm^3^	2–8	12 (4, 20) >300: *n* = 0 <300: *n* = 16	1586 (244, 2133) >300: *n* = 9 <300: *n* = 4	0.1
Neutrophils, %	3–5	47 (26, 62) >80: *n* = 0 <80: *n* = 16	91 (88, 93) >80: *n* = 11 <80: *n* = 2	**0.03**
Glucose, mmol/L	2.8–3.9	3.8 (2.9, 4.2) <2.7: *n* = 5 >2.7: *n* = 11	1.3 (0.5, 3.7) <2.7: *n* = 9 >2.7: *n* = 4	0.99
Protein, g/L	0.1–0.3	0.9 (0.5, 1.1) >1.0: *n* = 5 <1.0: *n* = 11	1.7 (1.0, 5.9) >1.0: *n* = 10 <1.0: *n* = 3	**0.02**
Lymphocytes, %	90–95	69 (39, 84)	6 (5, 9)	**<0.001**

* The differences are statistically significant in the case of a 2-tailed *p* < 0.05 (the Wald test) and are highlighted in bold.

**Table 3 metabolites-15-00527-t003:** Concentrations of aromatic metabolites in blood serum samples of healthy donors (*n* = 48) [28,29] and patients with long-term sequelae of severe brain damage, nmol/L.

Aromatic Metabolite	Serum Samples from Healthy Donors (*n* = 48)	Group I. Serum Samples (*n* = 16) from Patients Without Secondary Bacterial Meningitis	*p*-Value *	Group II. Serum Samples (*n* = 13) from Patients with Secondary Bacterial Meningitis	*p*-Value *
4-Hydroxyphenyllactic acid (*p*-HPhLA)	1212 (959, 1557)	975 (821, 1398)	0.89	1676 (1378, 2184)	**<0.001**
4-Hydroxybenzoic acid (*p*-HBA)	18 (16, 24)	12,020 (8408, 21,565)	**<0.001**	10,300 (7523, 19,520)	**<0.001**
4-Hydroxyphenylacetic acid (*p*-HPhAA)	316 (0, 461)	1238 (479, 1918)	**<0.001**	2636 (500, 5512)	**<0.001**
3-Phenylpropionic acid (PhPA)	458 (269, 724)	<250	-	<250	-
4-Hydroxyphenylpropionic acid (*p*-HPhPA)	9 (<7.5, 14)	36 (16, 58)	**<0.001**	<7.5 (<7.5, 10)	0.35
3-Phenyllactic acid (PhLA)	315 (249, 391)	2445 (1713, 3856)	**<0.001**	5602 (4610, 8132)	**<0.001**
5-Hydroxyindole-3-acetic acid (5HIAA)	78 (64, 93)	40 (20, 51)	0.43	71 (52, 94)	0.73
Indole-3-lactic acid (3ILA)	1068 (839, 1272)	595 (433, 676)	**<0.001**	1547 (957, 1996)	0.27
Indole-3-carboxylic acid (3ICA)	22 (18, 26)	33 (29, 39)	**<0.001**	31 (26, 42)	**<0.001**
Indole-3-acetic acid (3IAA)	1823 (1513, 2377)	928 (673, 1161)	**<0.001**	362 (200, 624)	**<0.001**
Indole-3-propionic acid (3IPA)	1362 (773, 2087)	<200 (<200, 245)	-	<200	-

Comparison of the groups by this variable was not performed due to a lack of detected values. * The differences between patients’ groups and healthy donors are statistically significant in the case of 2-tailed *p* < 0.05 (the Wald test with Bonferroni correction) and are highlighted in bold.

**Table 4 metabolites-15-00527-t004:** Concentrations of aromatic metabolites in the blood serum and cerebrospinal fluid samples of patients with long-term sequelae of severe brain damage, nmol/L.

Aromatic Metabolite	Biological Sample	Group I. Samples (*n* = 16) from Patients Without Secondary Bacterial Meningitis	Group II. Samples (*n* = 13) from Patients with Secondary Bacterial Meningitis	*p*-Value *
4-Hydroxyphenyllactic acid (*p*-HPhLA)	Serum	975 (821, 1398)	1676 (1378, 2184)	0.13
	CSF	415 (329, 658)	2578 (2410, 4111)	**<0.001**
4-Hydroxybenzoic acid (*p*-HBA)	Serum	12,020 (8408, 21,565)	10,300 (7523, 19,520)	0.76
	CSF	32 (26, 53)	42 (24, 99)	0.09
4-Hydroxyphenylacetic acid (*p*-HPhAA)	Serum	1238 (479, 1918)	2636 (500, 5512)	0.09
	CSF	141 (66, 280)	1827 (362, 3894)	**<0.001**
3-Phenylpropionic acid (PhPA)	Serum	<250	<250	-
	CSF	<25	<25	-
4-Hydroxyphenylpropionic acid (*p*-HPhPA)	Serum	36 (16, 58)	<7.5 (<7.5, 10)	**0.04**
	CSF	<7.5 (<7.5, 2)	<7.5	-
3-Phenyllactic acid (PhLA)	Serum	2445 (1713, 3856)	5602 (4610, 8132)	0.10
	CSF	50 (35, 69)	572 (431, 678)	**0.03**
5-Hydroxyindole-3-acetic acid (5HIAA)	Serum	40 (20, 51)	71 (52, 94)	**0.01**
	CSF	36 (20, 144)	124 (72, 259)	0.27
Indole-3-lactic acid (3ILA)	Serum	595 (433, 676)	1547 (957, 1996)	**<0.001**
	CSF	8 (3, 14)	156 (64, 476)	**0.02**
Indole-3-carboxylic acid (3ICA)	Serum	33 (29, 39)	31 (26, 42)	0.70
	CSF	11 (7, 13)	13 (12, 20)	0.12
Indole-3-acetic acid (3IAA)	Serum	928 (673, 1161)	362 (200, 624)	**0.04**
	CSF	28 (21, 51)	90 (37, 148)	0.05
Indole-3-propionic acid (3IPA)	Serum	<200 (<200, 245)	<200	0.38
	CSF	3 (2, 6)	4 (2, 4)	0.99

Comparison of the groups by this variable was not performed due to a lack of detected values. * The differences are statistically significant in the case of 2-tailed *p* < 0.05 (the Wald test with Bonferroni correction) and are highlighted in bold.

## Data Availability

The original contributions presented in this study are included in the article. Further inquiries can be directed to the corresponding author(s).

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
