# Peer review of "Differences in the Profile of Aromatic Metabolites in the Corresponding Blood Serum and Cerebrospinal Fluid Samples of Patients with Secondary Bacterial Meningitis"

_metabolites, 2025, doi:10.3390/metabo15080527_

Round 1

Reviewer 1 Report

Comments and Suggestions for Authors

The authors describe a study involving patients with long-term sequelae of severe brain damage. They compare two groups of patients with (group II) and without (group (I) secondary bacterial meningitis. They have 11 patients without- and 4 patients with secondary bacterial meningitis. However, they collected 16 CSF and serum pairs of samples from group I and 13 CSF and serum pairs of samples from group II. Ultimately, in both groups same patients provided multiple samples at different timepoints. As mentioned in the text, “two patients with meningitis had 2 points, one had 3 points and one had 6 points”. We recognize how difficult it can be to collect sufficient samples for such studies. However, collecting 13 pairs of samples from 4 patient in one group and presenting all 13 pairs of samples as if they were independent is not appropriate. Therefore, all presented statistics are meaningless. One could have accounted for the sample dependence using mixed-effects models. However, there is no hint in the text that such method like mixed-effect model was performed by the authors. Therefore, all presented stats are flawed and meaningless.

The observed (probably) valid differences in the concentrations of the metabolites of (p-HPhLA), PhLA, 5-hydroxyindole-3-acetic, and in-dole-3-lactic acids in serum and CSF are very important. However, the presentation in this form is not correct. If published, there is a good chance that the paper must be retracted later because proven otherwise. There are also many uncontrolled variables that could give rise to the reported measurement, they may not have anything to do with bacterial meningitis. For example, the 4 patients in group II are heavily burdened with other disease such diabetes mellitus, hypertension, and gastrointestinal disorders. Each one of them have the potential to heavily influence the measured concentrations of these metabolites.

Therefore, I have to reject the paper in present form. The editor should consider inviting the authors to perform appropriate data analysis and resubmit.

Author Response

Dear Reviewer,

We sincerely thank you for your time spent studying the article, understanding the complexity of obtaining such real samples and valuable comment.

We studied the proposed statistical method and applied it to recalculate all the results in Tables 2-4. It significantly affected the statistical data we obtained. To confirm the analysis performed, we attach the results of this statistical analysis in separate file. In addition, we approached the assessment of the obtained data more critically in general and, among other things, removed the results of model construction to the Supplementary Materials. In general, the text of the manuscript was significantly changed in accordance with the new statistical data and comments of other Reviewers. We hope that the amended version of the manuscript will satisfy you.

Reviewer 2 Report

Comments and Suggestions for Authors

The manuscript from Pautova et al studies the ratio of aromatic metabolites in the serum and cerebrospinal fluid (CSF) of patients with long-term sequelae of severe brain damage who are suspected of having secondary bacterial meningitis. For this study, which is a pilot given the limited number of patients with secondary bacterial meningitis, authors aim to explore wether these metabolites, particularly their ratios, can provide specific diagnostic information for secondary bacterial meningitis. The authors also explore the prognostic value of other common biochemical and haematological parameters. The manuscript is well written. The methodological approach is well-described. The results are interesting and informative for the medical and research community; they are clearly presented in a classical and comprehensive way. The limitations and implications of the results obtained are well discussed albeit often in a too lengthy way. The manuscript would benefit from a more focused discussion.

MINOR

-       As currently presented in Table 1, data from the “Results from microbiological analysis” section does not clearly allow distinction between microbiological species detected in each biological fluid. See for example blood and urine from Group 1: Klebisiella pneumonia pertains to blood or urine?

Author Response

 "The limitations and implications of the results obtained are well discussed albeit often in a too lengthy way. The manuscript would benefit from a more focused discussion."

We thank the reviewer for his attention and high assessment of our results.

Discussion Section was changed and these changes are presented on pages 22-24.

"MINOR

-       As currently presented in Table 1, data from the “Results from microbiological analysis” section does not clearly allow distinction between microbiological species detected in each biological fluid. See for example blood and urine from Group 1: Klebisiella pneumonia pertains to blood or urine?"

We thank the reviewer for this comment. For clarity, horizontal lines have been added to the data presented in Table 1 to clearly separate the microbiological results.

Reviewer 3 Report

Comments and Suggestions for Authors

The study is dedicated to the determination of aromatic metabolites in the serum and cerebrospinal fluid (CSF) in patients with long-term sequelae of severe brain damage and secondary bacterial meningitis. The manuscript corresponds to the scope of the journal and provides important findings for clinical laboratory diagnostics, demonstrating higher concentration of aromatic metabolites in in serum and CSF of patients with secondary bacterial meningitis. The study was well-designed, and the findings are important for clinical diagnostics and open prospects for further research into the topic. I would recommend acceptance after the revision. As for the revision, I would suggest first changing the title. It is too general and does not reflect the major findings of the study. All other points for revision are described below.

  1. In this sentence reference is required:

 Elevated CSF concentrations of 4-hydroxyphenyllactic acid (p-HPhLA) have been previously detected in post-neurosurgical patients with signs of secondary bacterial meningitis compared to those without signs of secondary bacterial meningitis.

  1. Is it not quite clear whether the ability of p-HPhLA to cross BBB was shown? If so, reference is required; if this is a hypothesis, comparison with metabolites of the same group and with similar MW is desirable.

We hypothesize that, on the one hand, p-HPhLA, the concentrations of which can be significantly increased in the serum of patients with infectious complications, has the ability to penetrate the BBB and lead to an increase in its CSF concentration.

  1. Please decipher CV

The CV for the determination of the analyte concentrations…

  1. For all tables: please, move the statistical description like

The differences are statistically significant in the case of 2-tailed p < 0.05 (the non-parametric Mann-Whitney U-test) and are highlighted in bold.

After the Tables. Leaving only the Table title before.

  1. The font in Figs. 5 and 6 can be enlarged; the signs are hardly visible.

  1. Discussion

It is not quite clear- healthy people or healthy people with a mutation of the gene? This part should be revised:

The serum 3IPA in constant low concentrations may also characterize healthy people. Dodd et al. demonstrated that mice with knockout gene fldC had low concentrations of propanoic acids (3IPA, PhPA, and p-HPhPA), which resulted in the elevated permeability of the gut barrier and impaired immune status as a hyperactivated response to microorganisms [21].

  1. In the following sentence changes can be suggested:

However, its concentrations in healthy donors were between groups I and II,

Concentrations in healthy donors were in the range between groups I and II,

  1. The same, changes can be applied here:

3ILA and 3IAA concentrations in serum independently showed good predictive ability for classification but not excellent (Table 5).

Clearer version:

3ILA and 3IAA concentrations in serum independently showed good, but not excellent predictive ability for classification (Table 5).

  1. In this sentence some elaboration will provide information for the reader:

5HIAA is a well-known serotonin metabolite and is not usually considered a marker of infectious complications [49].

How is it usually considered? How is it used in diagnostics?

  1. The following sentence has one extra quote, and probably informed consent also should be mentioned in Materials and Methods.

Informed Consent Statement: Informed consent was obtained from all subjects involved in the study.”

Should be changed to: Informed Consent Statement: Informed consent was obtained from all subjects involved in the study

Author Response

The study is dedicated to the determination of aromatic metabolites in the serum and cerebrospinal fluid (CSF) in patients with long-term sequelae of severe brain damage and secondary bacterial meningitis. The manuscript corresponds to the scope of the journal and provides important findings for clinical laboratory diagnostics, demonstrating higher concentration of aromatic metabolites in in serum and CSF of patients with secondary bacterial meningitis. The study was well-designed, and the findings are important for clinical diagnostics and open prospects for further research into the topic. I would recommend acceptance after the revision.

We thank the reviewer for his attention and high assessment of our results.

As for the revision, I would suggest first changing the title. It is too general and does not reflect the major findings of the study.

We thank the reviewer for this comment and suggest an alternative title for consideration: “Differences in the profile of aromatic metabolites in the corresponding blood serum and cerebrospinal fluid samples of patients with secondary bacterial meningitis”

All other points for revision are described below.

  1. In this sentence reference is required:

 Elevated CSF concentrations of 4-hydroxyphenyllactic acid (p-HPhLA) have been previously detected in post-neurosurgical patients with signs of secondary bacterial meningitis compared to those without signs of secondary bacterial meningitis.

We thank the reviewer for this comment and added the reference 25 here, however, the next sentence is corresponded to the same reference and has its number in the end.

  1. Is it not quite clear whether the ability of p-HPhLA to cross BBB was shown? If so, reference is required; if this is a hypothesis, comparison with metabolites of the same group and with similar MW is desirable.

We hypothesize that, on the one hand, p-HPhLA, the concentrations of which can be significantly increased in the serum of patients with infectious complications, has the ability to penetrate the BBB and lead to an increase in its CSF concentration.

We thank the reviewer for this comment. As noted at the beginning of the sentence, this is a hypothesis that we do not have the opportunity to test yet. However, we have added the information about the possible mechanisms of its transport through the BBB in the last paragraph of the Discussion: “The second part of the hypothesis about the p-HPhLA accumulation in the CSF included the possible p-HPhLA penetration from the bloodstream through the BBB. p-HPhLA is a hydrophilic compound with a hydroxyl and a carboxyl groups, and accordingly, low lipophilicity, which complicates passive diffusion through the lipid layer of the BBB membranes, and also has a relatively high molecular weight of 182 Da. The most possible ways of its penetration through the BBB are via 1) the LAT1 L-type amino acid transporter 1 that is used for tyrosine, phenylalanine and tyrosine transport [50]; and 2) OAT organic anion transporters [51].”.

  1. Please decipher CV

The CV for the determination of the analyte concentrations…

We thank the reviewer for this comment. However, CV (coefficient of variation) was decipher on page 3, the second paragraph; and at the end of the manuscript in the section Abbreviations.

  1. For all tables: please, move the statistical description like

The differences are statistically significant in the case of 2-tailed p < 0.05 (the non-parametric Mann-Whitney U-test) and are highlighted in bold.

After the Tables. Leaving only the Table title before.

We thank the reviewer for this comment. We moved the statistical description after the Tables.

  1. The font in Figs. 5 and 6 can be enlarged; the signs are hardly visible.

We thank the reviewer for this comment. Fig. 6 was moved into Supplementary 2 in a larger size. Fig 5 (now Fig. 1) was enlarged.

  1. Discussion

It is not quite clear- healthy people or healthy people with a mutation of the gene? This part should be revised:

The serum 3IPA in constant low concentrations may also characterize healthy people. Dodd et al. demonstrated that mice with knockout gene fldC had low concentrations of propanoic acids (3IPA, PhPA, and p-HPhPA), which resulted in the elevated permeability of the gut barrier and impaired immune status as a hyperactivated response to microorganisms [21].

We thank the reviewer for this comment. We have changed text: “The serum 3IPA in constant low concentrations may also characterize healthy people. Dodd et al. demonstrated that gnotobiotic mice colonized with C. sporogenes with knockout gene fldC had low concentrations of propanoic acids (3IPA, PhPA, and p-HPhPA). This fact resulted in the elevated permeability of the gut barrier and impaired immune status as a hyperactivated response to microorganisms”

  1. In the following sentence changes can be suggested:

However, its concentrations in healthy donors were between groups I and II,

Concentrations in healthy donors were in the range between groups I and II,

We thank the reviewer for this comment. The corresponding changes have been made.

  1. The same, changes can be applied here:

3ILA and 3IAA concentrations in serum independently showed good predictive ability for classification but not excellent (Table 5).

Clearer version:

3ILA and 3IAA concentrations in serum independently showed good, but not excellent predictive ability for classification (Table 5).

We thank the reviewer for this comment. We removed this sentence to reduce speculation about the statistical analysis of our very small patient sample.

  1. In this sentence some elaboration will provide information for the reader:

5HIAA is a well-known serotonin metabolite and is not usually considered a marker of infectious complications [49].

How is it usually considered? How is it used in diagnostics?

We thank the reviewer for this comment. We clarified the known diagnostics significance of 5HIAA: “5HIAA is a well-known serotonin metabolite and its serum concentration is a potential indicator for the diagnosis and monitoring of carcinoid tumors originating from enterochromaffin cells in the small intestine; hence, is not usually considered a marker of infectious complications [49].

  1. The following sentence has one extra quote, and probably informed consent also should be mentioned in Materials and Methods.

Informed Consent Statement: Informed consent was obtained from all subjects involved in the study.”

Should be changed to: Informed Consent Statement: Informed consent was obtained from all subjects involved in the study

We thank the reviewer for this comment. An extra quote was removed and this Statement was duplicated in Materials and Methods

Round 2

Reviewer 1 Report

Comments and Suggestions for Authors

Thank you for the revised manuscript and for applying appropriate statistical methods. The case numbers are very low, however the findings are important and should be made available to the broader scientific community.